

# Genome-based development of 15 microsatellite markers in fluorescent multiplexes for parentage testing in captive tigers

Xiao Zhao[1,2,3,4], Qiguan Qiu[5], Chang Li[1,4,6], Dongke Fu[2,3,4], Xuesong Hu[2,4], Shengjie Gao[2,4], Yugang Zhu[7], Haofang Mu[8], Runping Wang[9], Huanming Yang[1,4,10] and Bo Li[4]

[1] BGI Education Center, University of Chinese Academy of Sciences, Shenzhen, China
[2] Forensic Genomics International (FGI), BGI-Shenzhen, Shenzhen, China
[3] Shenzhen Key Laboratory of Forensics, BGI-Shenzhen, Shenzhen, China
[4] BGI-Shenzhen, Shenzhen, China
[5] Changsha Ecological Zoo, Changsha, China
[6] BGI-Qingdao, BGI-Shenzhen, Qingdao, China
[7] Changsha Sanzhen Tiger Park, Changsha, China
[8] Center of Forensic Sciences, BGI, Beijing, China
[9] BGI Shaanxi Xixian new area Institute of Forensic Science, Xi'an, China
[10] China National GeneBank, BGI-Shenzhen, Shenzhen, China

Corresponding author
Bo Li, libo@genomics.cn

## ABSTRACT

As one of the most endangered species, tiger (*Panthera tigris*) inbreeding has become an urgent issue to address. Using a microsatellite (short tandem repeat, STR) identification system, paternity testing may be helpful to avoid inbreeding in captive breeding programs. In this study, we developed a genome-based identification system named tiger pedigree identification multiplex system (TPI-plex). By analyzing the entire tiger genome, 139,967 STR loci were identified and 12.76% of these displayed three to six alleles among three re-sequenced individual tiger genomes. A total of 204 candidate STRs were identified and screened with a reference population containing 31 unrelated captive tigers. Of these, 15 loci were chosen for inclusion in the multiplex panel. The mean allele number and mean expected heterozygosity (He) were 7.3333 and 0.7789, respectively. The cumulative probability of exclusion (CPE) and total probability of discrimination power (TDP) reached 0.999999472 and 0.999999999999995, respectively. The results showed that the TPI-plex system can be applied in routine pedigree identification for captive tigers. We also added a sex identification marker named TAMEL into the TPI-plex for sex determination.

## INTRODUCTION

The tiger (*Panthera tigris*) was listed as an endangered (EN) species by the International Union for Conservation of Nature (IUCN) in 1986. As a recognized keystone species, tigers play a key role in maintaining healthy ecosystems (*Cho et al., 2013*). Unfortunately,

the rapid loss of tigers was remarkable. Fewer than 4,000 wild tigers survived in areas that occupied only 7% of their historical range and the tigers were divided into two currently recognized subspecies (*Kitchener et al., 2017*; *Wilting et al., 2015*). The World Wildlife Fund (WWF) worked in alliance with local governments and agencies to protect wild tiger populations. Moreover, there has been an effort to conserve tigers through captive breeding programs and the efforts appeared to have paid off. The captive tiger population has greatly outnumbered the wild ones since 2007 (*Luo et al., 2008*). In China, the number of captive tiger population has grown rapidly since 2002 and has reached between 5,000 and 6,000 in the last 2 years (https://eia-international.org/where-are-the-tigers/).

For EN animals, the real purpose of protective captivity should not only concentrate on population expansion but genetic diversity preservation (*Reed & Frankham, 2003*). Inbreeding led to reduced genetic diversity and had a deleterious effect on the biological fitness of the population (*Keller & Waller, 2002*; *Reed & Frankham, 2003*; *Ruiz-López et al., 2012*). Research showed that tigers are one of the most inbred animals in captivity (*Begany & Criscuolo, 2009*). The captive South China tiger population was found to be suffering from inbreeding depression and a decline in genetic diversity (*Xu, Fang & Li, 2007*). High levels of inbreeding brought stillbirths and high infant mortality and very low genetic diversity within the captive population of Asian lions (*Atkinson et al., 2018*). Having a better understanding of the mating system of a species is the foundation to a successful breeding and captive management (*Ferrie et al., 2013*). Mastering the parent-child relationship of captive tigers is highly recommended in avoiding captive tiger inbreeding. The breeding programs of many captive species managed by zoos or other organizations depend on studbooks to record individual pedigrees (*Ferrie et al., 2013*; *Jones et al., 2002*). Accurate and complete pedigree information is essential for effective pedigree analysis (*Ferrie et al., 2013*). However, the recorded data in the studbook may be missing or incorrect and the information in the studbook should be supported by genetic analysis (*Ferrie et al., 2013*; *Xu, Fang & Li, 2007*).

Microsatellites (short tandem repeats, STRs) proved to be one of the most powerful genetic markers for kinship analysis of animals and have been generally applied for this purpose (*Ichikawa et al., 2001*; *Luikart et al., 1999*; *Pei et al., 2018*; *Queller, Strassmann & Hughes, 1993*; *Webster & Reichart, 2005*). Isolating polymorphic microsatellites efficiently from the species genome is a crucial precondition of paternity testing for the proven STR method (*Webster & Reichart, 2005*). Until now, there have been two alternative sources of microsatellite loci for the STR-based method of tiger paternity identification. First, one may select a set of microsatellite primers derived from the domestic cat (*Felis catus*) (*Menotti-Raymond et al., 1999*) to amplify the tiger's microsatellite DNA (*Wu et al., 2011*; *Zhang et al., 2003*). These investigations introduced an initial single-locus amplification in the target DNA and failed to provide the accuracy of paternity testing by calculating the cumulative probability of exclusion (CPE). Second, one may isolate microsatellite loci from the tiger genome (*Sharma et al., 2008*; *Williamson et al., 2002*; *Wu et al., 2009*; *Zhang et al., 2006a*, *2006b*). These investigations preselected specific repeated motifs as probes without knowing their abundance in the tiger genome and therefore inevitably introduced biases and limited the microsatellite types into

a small fixed subset (*Castoe et al., 2010*; *Malausa et al., 2011*). These methods are both quite labor-intensive.

Therefore, establishing a more efficient method for screening polymorphic loci unbiasedly from all types of microsatellite loci presenting in the tiger genome is essential for establishing a paternity test. Here, we developed a multiplex system in a single reaction tube that can serve as a convenient, effective and accurate method for paternity testing in the captive tigers. In this study, we screened highly polymorphic microsatellite loci on a genome-wide scale using bioinformatics analysis. We also validated the tiger amelogenin locus based on homology analysis for sex identification. We incorporated all the selected autosomal STR loci and the sex determination locus into one polymerase chain reaction (PCR) to establish an STR five-color fluorescent-multiplex system for simple and effective use. We used a reference population that consists of a group of unrelated individuals to investigate alleles, allelic frequencies, genotypes, genotype frequencies of each STR locus and assessed the validity and accuracy of the multiplex system for paternity testing and individualization in captive tigers.

## MATERIALS AND METHODS

### Sample collection

A total of 42 captive continental tigers were sampled (Table S1). 20 blood samples were collected from individuals T01–T20 via the femoral vein after anesthetization and 22 hair samples were collected from individuals RT01–RT22 (Table S1). Individuals T01–T20 are from Changsha Ecological Zoo. We want to identify their parent–child relationships. Individuals T12–T20 are parents and do not have blood relationship, but they may mate and produce offspring T01–T11. There into, T01, T02, T06, T07, T08 and T09 six tigers were born from the same womb and T03, T04 and T05 three tigers were from the same womb. However, the biological parents were in doubt. The available recorded information was provided by the zoo staff (Table S2). Individuals RT01 to RT22 are other 22 unrelated tigers from different places. The study was approved by the *Institutional Review Board on Bioethics and Biosafety of BGI* (FT 16084).

### DNA isolation

Genomic DNAs from whole blood were extracted by TIANamp Blood DNA Kits® (TIANGEN Biotech, Beijing, China), following the manufacturer's instructions. The hair shaft of each hair sample was cut off and the remaining part containing the hair follicle was placed into a 1.5 ml Eppendorf tube and washed with double distilled water and absolute ethyl alcohol, respectively and digested by proteinase K. The genomic DNAs from the hair samples were isolated using Chelex 100.

### Selection of tiger markers

We downloaded the tiger reference genome (GCA_000464555.1 PanTig1.0) from the NCBI database and used Tandem Repeats Finder (v4.09) (*Benson, 1999*) for annotating the STR loci in the reference genome. We also downloaded the re-sequenced data from three tigers from the NCBI database (SRR640236, SRR640237 and SRR640238)

(*Xu et al., 2013*) and filtered the raw data by SOAPnuke (v2.0) (*Chen et al., 2017*). Based on the annotation results, we typed the STRs of the three tigers using lobSTR (*Gymrek et al., 2012*). We screened all of the valid STR loci with high polymorphisms that satisfied the following criteria: (1) the locus is a tetra-nucleotide or penta-nucleotide repeat, (2) the repeat units repeat 10 to 20 times in the reference genome and (3) three to six alleles in three re-sequenced individuals exist at the site.

To identify a tiger's sex, we validated the amelogenin locus based on homology analysis. We downloaded human amelogenin sequences (AMELX and AMELY) from the NCBI database, aligned them to the tiger genome, and found two homologous sequences in the tiger. Clustal X (v2.1) (*Jeanmougin et al., 1998*) was used to find the deletion polymorphism of amelogenin sequences.

Primer pairs were designed in flanking regions using OLIGO 7 (v7.56) (*Rychlik, 2007*) and the size range of the amplification products was controlled between 100 and 500 bp. The specificity of the primers was validated by PCR, and loci that could be easily amplified were reserved. The forward primers of the normal primer pairs were then labeled with different fluorescent-dye. Loci with inefficient amplification or split peaks were excluded.

## Multiplex amplification assay

The selected loci were incorporated into a multiplex amplification system. For the loci whose primers were labeled with the same dye color, their allele size ranges did not overlap each other. All of the fluorescence-labeled primer pairs were mixed in proportions that were determined from the results of multiple experiments guaranteeing good amplification at each site and peak height during capillary electrophoretic separation. The final primer concentration of each primer pair was in the range of 0.2–1.3 μM/μL. The primer mixture was used to amplify genomic DNAs by PCR and amplification was performed in a 10 μL reaction volume. The reaction mixture contained 50 mM KCl, 10 mM Tris–HCl (pH 8.3, 25 °C), 2.0 mM $MgCl_2$, 0.1 mg/ml BSA, 0.2 mM dNTPMix (dATP, dTTP, dCTP and dGTP mixed equally), 0.2 units DNA polymerase (EzAmp® Fast Taq DNA Polymerase) and 0.1–2 ng genomic DNA. For each reaction, the PCR conditions were as follows: 1 cycle of 95 °C for 5 min, 30 cycles of 95 °C for 10 s, 58 °C for 1 min, 70 °C for 20 s, and 1 cycle of 60 °C for 1 h with Applied Biosystems® Veriti® Thermal Cycler.

## Electrophoresis separation and data analysis

The PCR products (1 μl) were mixed with loading buffer composed of Hi-DiTM formamide and internal size standard Salmon 500 Plus at a 9:0.3 (v/v) ratio. The electrophoretic separations were performed on an ABI 3500 Genetic Analyzer (Thermo Fisher, Waltham, MA, USA). Then the collected data were analyzed with GeneMapper® ID-X Software Version 1.5.

## Allele sequencing and genotyping

A homozygote was selected at each locus and the amplification products were sent to Sangon Biotech (Sangon, Shanghai, China) for Sanger sequencing in both forward and reverse
directions after performing agarose gel electrophoresis for validation. The sequenced alleles were named by the number of repetitions of the motif according to the nomenclature of STR recommended by the International Society for forensic genetics (ISFG) (*Mayr, 1995*; *Olaisen et al., 1998*). The repetition numbers of the motifs of unsequenced alleles were deduced in reference to the repeat sequence structures of both the sequenced alleles and the reference genome and the observed sizes of both sequenced and unsequenced alleles. The panel and bin files were programed for the GeneMapper® ID-X Software Version 1.5 to genotyping analysis. The allelic ladders of each locus were listed in Table S3.

## Population genetic analysis

From the 42 captive continental tigers, a total of 31 unrelated tigers (T12–T20 and RT01–RT22) were selected as the reference population for population genetic analyses of the 15 autosomal STR loci. The Hardy–Weinberg equilibrium (HWE) for each locus was tested by using Genepop v4.7 (*Raymond & Rousset, 1995*; *Rousset, 2008*). Expected heterozygosity ($H_e$: Eq. (1)), probability of exclusion (PE: Eq. (2)), and discrimination power (DP: Eq. (3)) of each locus, cumulative probability of exclusion (CPE: Eq. (4)), and cumulative discrimination power (TDP: Eq.(5)) were calculated according to the formulas below (*Frankham, Ballou & Briscoe, 2010*; *Kloosterman, Budowle & Daselaar, 1993*; *Li et al., 2018*; *Wu et al., 2009*; *YiPing, 2016*).

$$H_e = 1 - \sum_{i=1}^{k} p_i^2 \tag{1}$$

$k$ is the number of alleles and $p_i$ the allele frequency of the $i$th allele at the target locus

$$PE = \sum_{i=1}^{k} p_i(1 - p_i)^2 - 1/2 \sum_{\substack{i=1 \\ i \neq j}}^{k-1} \sum_{j=i+1}^{k} p_i^2 p_j^2 (4 - 3p_i - 3p_j) \tag{2}$$

$p_i$ and $p_j$ are respectively the allele frequency of the $i$th and $j$th allele at the target locus, with $i$ not equal to $j$

$$CPE = 1 - \prod_{i=1}^{k} (1 - PE_k) \tag{3}$$

$PE_k$ is PE for each of $k$ loci

$$DP = 1 - \sum_{i=1}^{k} g_i^2 \tag{4}$$

$g_i$ is the frequency of each genotype

$$TDP = 1 - \prod_{i=1}^{k} (1 - DP_k) \tag{5}$$

$DP_k$ is DP for locus $k$

## Sensitivity testing

To evaluate the sensitivity of the TPI-plex amplification system, the DNA from the individual of T08 was chosen as the control DNA and used to perform the sensitivity testing experiment. A series of template DNA quantities were diluted in a 10 μL PCR reaction system and the DNA concentrations, from high to low, were as follows: 2 ng 10 μL$^{-1}$, 1 ng 10 μL$^{-1}$, 0.5 ng 10 μL$^{-1}$, 0.25 ng 10 μL$^{-1}$, 0.125 ng 10 μL$^{-1}$, 0.0625 ng 10 μL$^{-1}$ and 0.03125 ng 10 μL$^{-1}$. Each quantity of DNA was analyzed in triplicate and a negative control group was set up.

## Specificity testing

As DNA from human or other non-human species may be mix in detected material, the TPI-plex amplification system was tested with DNA from a range of species including human, sheep, chick, duck, dog and rat, under the same PCR amplification condition to estimate potential interference. Genomic DNAs from human and dog samples were extracted from whole blood with TIANamp Blood DNA Kits® (TIANGEN Biotech, Beijing, China), while others were extracted from fresh tissue (purchased from markets) using TIANamp Genomics DNA Kits® (TIANGEN Biotech, Beijing, China). Each DNA was analyzed in duplicate and a negative control group was set up.

# RESULTS AND DISCUSSION

## Establishment of TPI-plex identification system

### Selection of tiger markers

Microsatellites have a high mutation rate as compared to other types of known genetic polymorphisms (*Ballantyne et al., 2010*; *Webster & Reichart, 2005*). Microsatellites analyses are based on simple PCR, which allows for typing in samples of low DNA quality or concentration (*Dawson et al., 2013*; *Webster & Reichart, 2005*). Therefore, microsatellites are the most commonly used genetic marker in parentage identification of animals (*Vignal et al., 2002*; *Webster & Reichart, 2005*), the paternity testing in tigers is no exception (*Wu et al., 2011*; *Zhang et al., 2003*). A total of 139,967 valid STR loci were identified across the whole tiger genome and 31.48% and 12.76% of these STR loci display two and three to six alleles (Fig. S1A), respectively. As the tiger genome sequence shows a 95.6% similarity to that of the domestic cat (*Cho et al., 2013*), we mapped all the detected STR loci from the tiger genome onto a cat karyotype, based on their genomic homology, to show their distribution on chromosomes (Fig. S1B). We generated 204 candidate STR loci (Table S4) of which 84.11% displayed three alleles and covered nineteen chromosomes of the tiger (Fig. 1). Then, primers were designed for 49 STR loci of 204 candidate STR loci based on the tiger reference and 27 pairs of primers on 27 STR loci were effective. Three loci were excluded because of weak or split peaks after the detection of fluorescent-labeled PCR products. The remaining 24 loci were distributed on 14 chromosomes and were named as DA1S1290, DA1S1470, DA2S1059, DA2S1575, DA3S1145, DA3S461, DA3S1123, DB1S1259, DB1S1096, DB1S542, DB2S734, DB2S23, DB3S187, DB4S1505, DB4S2706, DB4S2753, DC1S1364, DD2S793, DD3S899, DD3S86, DD4S705, DE1S613, DF1S579 and DF2S497, respectively (Fig. 1).
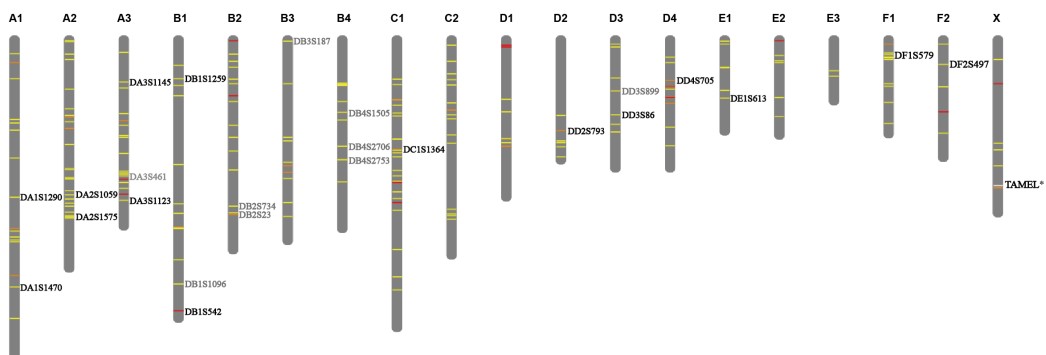

**Figure 1 Distribution of selected STRs on chromosomes.** The 204 candidate STRs are labeled on chromosome using different colors based on the allele polymorphism. Yellow, orange and red represent the allele number is 3, 4 and 5, respectively. The 24 validated loci were denoted, among which fifteen retained loci are denoted with black fonts, nine excluded loci are denoted with gray fonts. The position of sex identification locus TAMEL is marked with white line on chromosome *X* and denoted with black fonts.

In humans, sex identification in forensic multiplexes is based upon the amelogenin gene on both the *X* and *Y* chromosomes, which is commonly used in sex genotyping (*Akane et al., 1991*; *Nakahori, Takenaka & Nakagome, 1991*). Similarly, we discovered two sequences of the tiger amelogenin gene on scaffolds of ATCQ01070658.1 and ATCQ01738.1 by homology searching (Fig. S2). Interestingly, on the tiger amelogenin locus, we found a deletion polymorphism (20 bp) which may be used to identify the tiger's sex. We named the tiger amelogenin locus *TAMEL*, amelogenin *X*-linked as *TAMELX* on ATCQ01070658.1 and amelogenin *Y*-linked as *TAMELY* on ATCQ01738.1, respectively.

### *TPI-plex identification system development*

We established an STR five-color fluorescent-multiplex system, with 15 autosomal STRs loci and a sex identification locus (Table 1). The 16 loci (DA1S1290, DA1S1470, DA2S1059, DA2S1575, DA3S1123, DA3S1145, DB1S542, DB1S1259, DD2S793, DD3S86, DD4S705, DE1S613, DF2S497, DF1S579 and TAMEL; Fig. 1) are distributed on 12 chromosomes and the 16-plex identification system used fluorescent forward primers labeled at the 5′ end with blue (6′-FAM), green (HEX), yellow (TAMRA), or red (ROX) dyes (Table 1).

To characterize each locus, we sequenced PCR amplification products from homozygotes (*Xu et al., 2005*) and sequencing results (Table S5) provided the repeat numbers of motif units (Table S6). Sequenced alleles were defined on the basis of the nomenclature of STRs (*Mayr, 1995*). Most of these loci are simple repeats of tetra-nucleotide or penta-nucleotide motifs (Table S6). DE1S613, DF1S579, DA3S1145 and DD2S793 are compound repeats due to their two different types of motifs. DB1S1259 and DD2S705 are complex repeats composed of tetra- and a few penta-nucleotide motifs. Their alleles were designated using the method which assumed the region was a general tetrameric repeat structure (*Hellmann et al., 2006*). The amplified fragments of both DB1S1259 and DD4S705 loci consist of a complex hypervariable region mainly based

**Table 1 The information of selected loci in the multiplex system and primer sequences of each locus.**

| Locus | Scaffold | Position on scaffold | Repeat motif | Size range (bp) | Dye | Primer sequences (5′–3′) |
|---|---|---|---|---|---|---|
| TAMEL | – | – | – | 80–115 | 6′-FAM | GAGAGGCCAARTAGGAGTGTGC TTCAAGATGTTTCTCAGTCC |
| DA3S1123 | scaffold94140 | 28,202 | AAAAC | 206–276 | 6′-FAM | ATGTATGTCTCCTGCACATGCTTCCAC CTGTCATGGATAATGTGCTTGAGTCCCT |
| DA2S1059 | scaffold67413 | 66,413 | AGAT | 332–380 | 6′-FAM | CCTTGTACGAAAACAGGCAGTAAGCCA AGGCAAGACATTCACTTCTTAGGCA |
| DB1S1259 | scaffold87533 | 4,877 | AAAG | 388–448 | 6′-FAM | TCCTCTTCTGGTGGGAACTTCTC ATTTTGCAATTTCTTAGCATAGTGGC |
| DD3S86 | scaffold62023 | 14,317 | AAAG | 157–205 | HEX | TCCTTTTGGGATTTAGTACACCAG ATTTTAAAGTCAACACCTGTA |
| DE1S613 | scaffold91793 | 6,047 | AAGG | 222–270 | HEX | GCCCACCACCTTCAAAATAGCCAT CCTCTTGACAGCCCTCCTTATACCTC |
| DF1S579 | scaffold79 | 17,437 | AAAG | 307–355 | HEX | CCCTTGCTTTTAATGAGGCATAACCTT GCTTCCCACTCCCTATGTATGGC |
| DA2S1575 | scaffold88063 | 10,431 | AAAAG | 368–428 | HEX | TATGCTTAGGTCTGCTCATCAAGGG GATGATCATAAGCGGAAATGCACGAG |
| DF2S497 | scaffold72035 | 19,788 | AAAG | 132–178 | TAMRA | CACTGGTATGTTTAAGGGAATGTCA CACATAGCACATCATCTTAACACA |
| DA3S1145 | scaffold74228 | 13,337 | AAAG | 215–263 | TAMRA | TGGGGTAGTCGTTATACAACCGAA CTTAACATCAGTGAGCCCAATGTGGA |
| DD2S793 | scaffold140970 | 2,179 | AGAT | 317–365 | TAMRA | AACATAAGTGGGAAGCTGGTATCTGA TCTCCCCTTAGCTGTTATGTGCAGT |
| DD4S705 | scaffold96585 | 34,146 | AAAG | 377–425 | TAMRA | GTTGCTAAGGCTCACCCATACAAA CAATAAAGAGCCCTTTGATCAGTCC |
| DB1S542 | scaffold96253 | 30,364 | AAGG | 159–207 | ROX | CGGTTCCTGTTACACTTCTTAGCCTT AGGTACAATAATATTACACGGAAAGCA |
| DA1S1290 | scaffold89346 | 24,116 | AAAG | 212–272 | ROX | CTGAGTCTTCAAGCTGGGTTATCACA ATACACAGCTTCTCAAATGCCATCC |
| DA1S1470 | scaffold74155 | 15,379 | AAAG | 314–362 | ROX | CACACACCACAGAGCACTAGGCA ATCTGGTCTGGGTCTTTTAACTCCT |
| DC1S1364 | scaffold80644 | 19,735 | AAAG | 378–426 | ROX | CTGCCATAATCCAGATGTGTAAACCA AACAAATCTCAAACATTTCGGCTCT |

on a tetrameric motif of (AAAG) repeat and a pentameric motif of (AAAAG) repeat. The nomenclature was based on assuming a general tetrameric repeat structure (Table S6). Eleven out of the fifteen loci had the same repeat sequences as the reference genome, while four loci (DB1S1259, DE1S613, DF1S579 and DA3S1145) displayed polymorphisms in repeat sequences.

## Evaluation of the TPI-plex identification system
### The CPE and TDP of TPI-plex system

To assess the TPI-plex, we calculated the allele frequencies (Table 2) and genotype frequencies (Table S7) in the reference population ($n = 31$). The repetition numbers of the motifs of unsequenced alleles were deduced (see the method "Allele Sequencing and

**Table 2 Alleles and allelic frequencies of the 15 microsatellite loci in the reference population (*n* = 31).**

| Locus | Alleles and allelic frequency | | | | | | | | | | |
|---|---|---|---|---|---|---|---|---|---|---|---|
| DA3S1123 | Allele | 11.2 | 12.2 | 13.2 | 14.2 | 15.2 | | | | | |
| | Frequency | 0.2097 | 0.2581 | 0.4194 | 0.0161 | 0.0968 | | | | | |
| DA2S1059 | Allele | 8 | 9 | 10 | 11 | 12 | | | | | |
| | Frequency | 0.4355 | 0.3065 | 0.1452 | 0.0323 | 0.0806 | | | | | |
| DB1S1259 | Allele | 18.1 | 19.2 | 20.1 | 20.2 | 21.1 | 22.1 | 22.2 | 23.2 | 25.1 | 25.3 | 26.3 |
| | Frequency | 0.0484 | 0.1290 | 0.1452 | 0.0806 | 0.1129 | 0.1613 | 0.0484 | 0.0323 | 0.0484 | 0.0968 | 0.0968 |
| DD3S86 | Allele | 13 | 14 | 15 | 16 | 17 | 18 | 19 | | | | |
| | Frequency | 0.0323 | 0.4516 | 0.0484 | 0.0806 | 0.2097 | 0.0968 | 0.0806 | | | | |
| DE1S613 | Allele | 11 | 12 | 13.2 | 14.2 | 14.3 | 15.2 | 15.3 | 16.2 | 17 | 18 | 19.1 |
| | Frequency | 0.0968 | 0.0161 | 0.1613 | 0.0161 | 0.1129 | 0.0161 | 0.0806 | 0.0323 | 0.0161 | 0.1774 | 0.2741 |
| DF1S579 | Allele | 9.1 | 11.1 | 12.1 | 13.1 | 14.1 | 15 | 16 | 17 | 18 | | |
| | Frequency | 0.1935 | 0.1129 | 0.1774 | 0.1129 | 0.0968 | 0.1613 | 0.0323 | 0.0645 | 0.0484 | | |
| DA2S1575 | Allele | 7.1 | 8.1 | 9.1 | 10 | 11 | 12 | 13 | 14 | | | |
| | Frequency | 0.0161 | 0.1935 | 0.0806 | 0.1613 | 0.0968 | 0.3065 | 0.1129 | 0.0323 | | | |
| DF2S497 | Allele | 10 | 11 | 12 | 13 | 14 | | | | | | |
| | Frequency | 0.4516 | 0.2742 | 0.1613 | 0.0968 | 0.0161 | | | | | | |
| DA3S1145 | Allele | 15 | 16 | 17 | 18 | 19 | | | | | | |
| | Frequency | 0.2258 | 0.3387 | 0.2097 | 0.0806 | 0.1452 | | | | | | |
| DD2S793 | Allele | 9 | 10 | 11 | 12 | 13 | | | | | | |
| | Frequency | 0.1613 | 0.3226 | 0.2258 | 0.0645 | 0.2258 | | | | | | |
| DD4S705 | Allele | 17.3 | 18 | 18.2 | 19.2 | 19.3 | 20.2 | 20.3 | 22 | 22.2 | 22.3 | 24.3 |
| | Frequency | 0.1290 | 0.1452 | 0.0645 | 0.0645 | 0.1935 | 0.0645 | 0.1452 | 0.0968 | 0.0161 | 0.0323 | 0.0484 |
| DB1S542 | Allele | 11 | 12 | 13 | 14 | 15 | 16 | | | | | |
| | Frequency | 0.0161 | 0.4355 | 0.0161 | 0.2742 | 0.1613 | 0.0968 | | | | | |
| DA1S1290 | Allele | 7 | 10 | 11 | 12 | 13 | 14 | 15 | | | | |
| | Frequency | 0.0806 | 0.1774 | 0.0645 | 0.1935 | 0.2419 | 0.1129 | 0.1290 | | | | |
| DA1S1470 | Allele | 9 | 10 | 11 | 12 | 13 | 14 | 15 | | | | |
| | Frequency | 0.3387 | 0.2258 | 0.0161 | 0.0484 | 0.1290 | 0.1774 | 0.0645 | | | | |
| DC1S1364 | Allele | 12.2 | 13.2 | 14 | 14.2 | 15.2 | 16.2 | 17.2 | 18.2 | | | |
| | Frequency | 0.1613 | 0.0645 | 0.0161 | 0.0161 | 0.2097 | 0.3871 | 0.1129 | 0.0323 | | | |

Genotyping"). We provided the peak ratio of each locus for correct allele calling (Table S6). We ran this TPI-plex for each individual in the reference population (*n* = 31), and found that all of the STRs are polymorphic (Table 2; Table S7) and allele number ranged from 5 to 11. The observed and expected genotype distribution of 15 autosomal STR loci was listed in Table S8. DA3S1145, DA3S1123, DD3S86, DA1S1470, DA2S1059, DB1S1259 and DF1S579 displayed a departure from HWE (*P* < 0.05, Table 3). The deviation from HWE in allele frequencies at these microsatellite loci can be caused by natural selection, inbreeding, human disturbance, population degradation and small population size (*Antoro, Na-Nakorn & Koedprang, 2006*; *León, Chikhi & Bonhomme, 1997*; *Rousset & Raymond, 1995*; *Schaid et al., 2006*; *Yu et al., 2002*).

**Table 3 Characterization of 15 STR loci in the reference population ($n$ = 31).**

| Locus | No. of alleles | $H_e$ | HWE | PE | CPE | DP | TDP |
|---|---|---|---|---|---|---|---|
| DA3S1123 | 5 | 0.7040 | D[a] | 0.4818 | 0.4818 | 0.8304 | 0.8304 |
| DA2S1059 | 5 | 0.6878 | D | 0.4646 | 0.7226 | 0.8387 | 0.9726 |
| DB1S1259 | 11 | 0.8902 | D | 0.7856 | 0.9405 | 0.9282 | 0.99804 |
| DD3S86 | 7 | 0.7263 | D | 0.5385 | 0.9726 | 0.8637 | 0.999732 |
| DE1S613 | 11 | 0.8366 | NS[b] | 0.6929 | 0.9916 | 0.9303 | 0.9999813 |
| DF1S579 | 9 | 0.8626 | D | 0.7346 | 0.99776 | 0.9157 | 0.99999843 |
| DA2S1575 | 8 | 0.8127 | NS | 0.6521 | 0.999222 | 0.9136 | 0.999999864 |
| DF2S497 | 5 | 0.6852 | NS | 0.4622 | 0.999581 | 0.8262 | 0.9999999764 |
| DA3S1145 | 5 | 0.7627 | D | 0.5648 | 0.999818 | 0.8533 | 0.99999999653 |
| DD2S793 | 5 | 0.7638 | NS | 0.5640 | 0.9999206 | 0.8783 | 0.999999999578 |
| DD4S705 | 11 | 0.8783 | NS | 0.7643 | 0.9999813 | 0.9417 | 0.9999999999754 |
| DB1S542 | 6 | 0.6993 | NS | 0.4817 | 0.99999029 | 0.8345 | 0.99999999999593 |
| DA1S1290 | 7 | 0.8325 | NS | 0.6804 | 0.99999690 | 0.9282 | 0.999999999999707 |
| DA1S1470 | 7 | 0.7794 | D | 0.5971 | 0.99999875 | 0.8553 | 0.999999999999958 |
| DC1S1364 | 8 | 0.7617 | NS | 0.5774 | 0.999999472 | 0.8741 | 0.999999999999995 |
| | Mean: 7.3333 | Mean: 0.7789 | | Mean: 0.6028 | | Mean: 0.8808 | |

**Notes:**
[a] D, departure; from HWE, $P < 0.05$.
[b] NS, not significant; $P > 0.05$.

We evaluated the TPI-plex power in individualization and paternity testing. The expected heterozygosity ranged from 0.6852 to 0.8902 and the average $H_e$ was 0.7789. The PE of each locus ranged from 0.4183 to 0.8183, average PE being 0.6028 and the CPE reached 0.999999472 (Table 3). The DP ranged from 0.8262 to 0.9417 with an average of 0.8808. The TDP of all the 15 loci reached 0.99999999999995 (Table 3). In a previous study using 21 microsatellite loci and more than one multiplex panel for the Amur tiger, the CPE reached 0.9999 (Wu et al., 2009). Compared with this report, our method greatly simplifies this process and is more efficient.

### Sensitivity testing

Sensitivity testing can be used to find the DNA template usage limitation of the multiplex typing assay (Zhang et al., 2015). Here, we used DNA from the individual of T08 as control DNA. We performed the typing assay in triplicate in the range of total input DNA (0.03125–2 ng 10 μL$^{-1}$) under the same PCR conditions. We calculated the mean percentage of detected sites in the sensitivity testing and found that all 16 loci can be detected with DNA from 2 ng down to 0.5 ng. When the DNA amount were 0.25, 0.125, 0.0625 and 0.03125 ng, the mean percentages of detected loci were 93.75%, 93.75%, 37.5% and 18.75%, respectively. When the DNA template amount decreased to 0.125 ng, only one loci could not be detected. Furthermore, we calculated the mean peak height in the sensitivity testing. When the DNA template amount varied from 2 ng down to 0.5 ng, the mean peak height was from 10,950.37 relative fluorescence units (RFU) down to 5,752.40 RFU. In summary, the minimal DNA template was 0.5 ng for the TPI-plex.

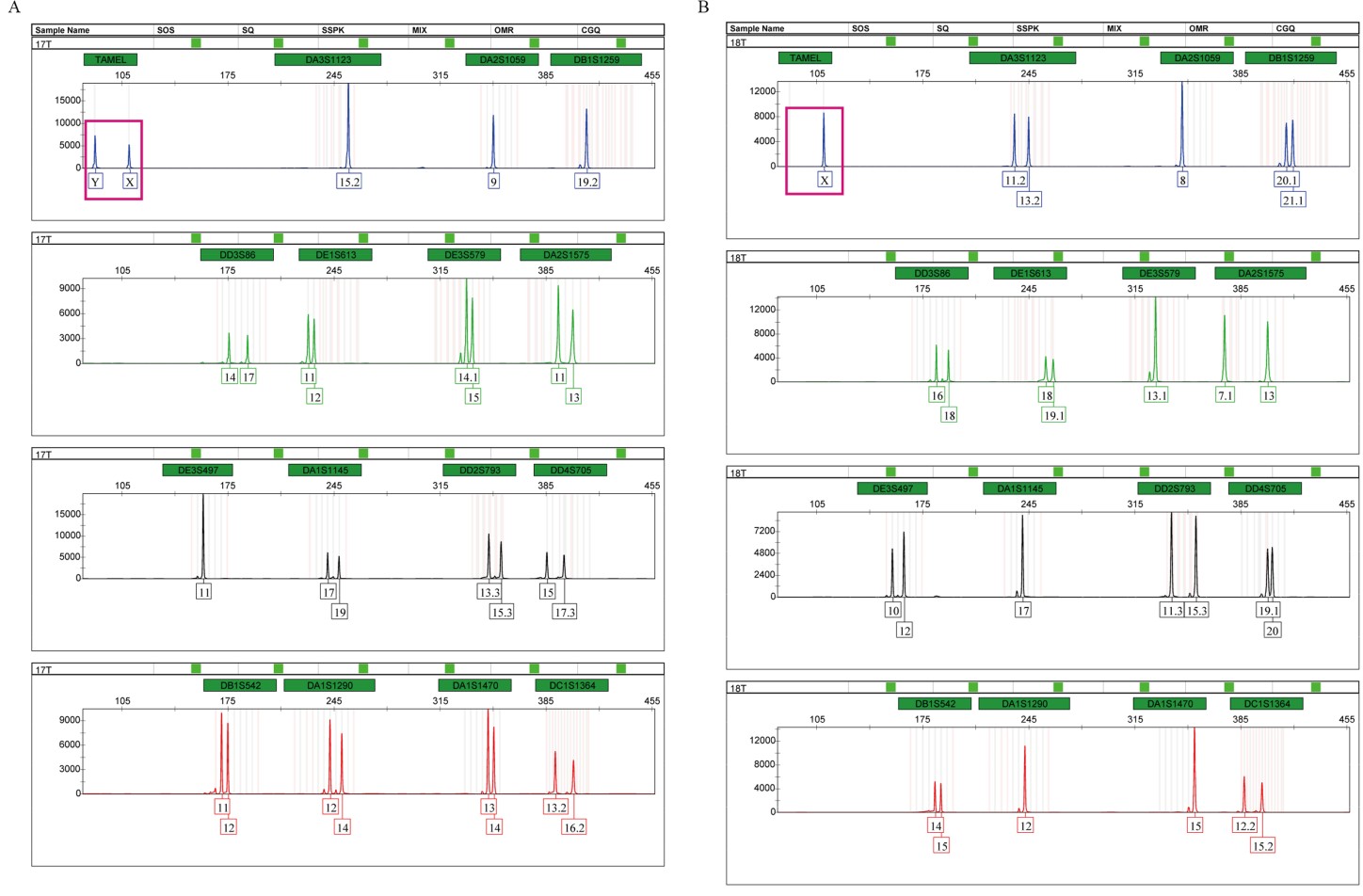

**Figure 2 Amplification products of a male (A) and a female (B), detected by an ABI 3500 genetic analyzer.**

### Species specificity

We performed the TPI-plex on the common species of human, sheep, chick, duck, dog and rat and showed that there was an off-range peak 180 to 210 bases long at DA2S1059 in human (193.0 bases), sheep (193.2 bases), dog (207.9 bases) and rat (182.9 bases). We detected a second off-range peak at DC1S1364 in sheep (357.0 bases) and dog (372.6 bases).

## Applications of the TPI-plex identification system

### Sex determination

Tigers have a chromosomal *XY* sex-determination system. The sex of tigers can be identified by analyzing the amplification products of the sex identification marker, TAMEL. We added the sex identification marker to the multiplex system for the first time and designed paired primers at both ends of the 20 bp deletion sequences. At the TAMEL locus, a male and female tiger display heterozygous (87 bp and 109 bp, namely, *Y* and *X*, Fig. 2A, boxed in carmine) and homozygous bands (109 bp, namely, *X*, Fig. 2B, boxed in carmine), respectively. Furthermore, we validated this sex marker by identifying the sexes of a group of tigers with known sex information (Table S1: RT01,

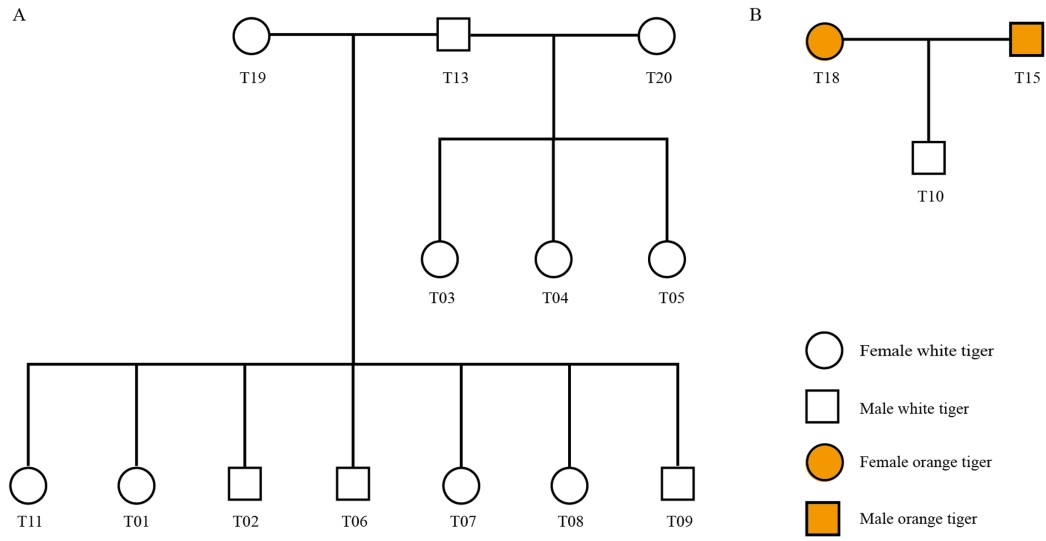

**Figure 3  Pedigree tree of the identified tigers.** (A) Pedigree relationship among 13 white tigers. (B) Pedigree relationship among T15, T18 and T10.

RT04, RT05, RT06, RT07, RT08, RT10, RT11, RT14 and RT15) using the TPI-plex system and compared our detected sex with information from these tiger's breeders. We found that our detection on all individuals was correct (Fig. S3), suggesting that the TAMEL locus can identify the sex correctly.

### Tiger pedigree identification

Using STR genotyping results, we determined the parent–child relationship among the 20 tigers from Changsha Ecological Zoo and validated the pedigree reconstruction using information provided by the zoo. As shown in Fig. 3, there were three families in all (Family 1: T13, T19, T01, T02, T06, T07, T08 and T09, Fig. 3A; Family 2: T13, T20, T03, T04 and T05, Fig. 3A; and Family 3: T15, T18 and T10, Fig. 3B) and T13 connects two families as a common father. Another 4 individuals, T12, T14, T16 and T17, have no blood relationships with the others. Table S9 lists genotyping information of the three families.

  We checked information from the zoo and found that T01, T02, T06, T07, T08 and T09 tigers were born from the same womb (Table S2). These six individuals and T11 shared the same dam, T19. Our genotyping results show that T19 was indeed biological mother of the seven tigers, since one of two alleles at each locus in the seven offspring individuals inherited from T19, which was in accordance with Mendelian inheritance. In addition, T03, T04 and T05 were three tigers born in a single birth.

  Interestingly, zoo staffs cannot confirm whether the white tiger T10 is from orange tiger family (T16 and T14) or from white tiger family (T18 and T15). Our identification results showed that T15 and T18 were biological parents of T10.

## CONCLUSIONS

We used the bioinformatics analysis method to identify tiger microsatellite loci on a genome-wide scale for the first time and screened 15 highly polymorphic microsatellite

loci distributed on 11 chromosomes from the tiger genome. We obtained a sex determination locus by validating the tiger amelogenin locus based on homology analysis. The 15 loci together with the sex determination locus were incorporated into one PCR reaction and a STR five-color fluorescent-multiplex system named TPI-plex was established. The TPI-plex system's CPE and TDP reached 0.999999472 and 0.999999999999995, respectively, suggesting that this TPI-plex can be applied for routine pedigree identification and individualization for tigers. The sex identification locus provided sex information of individuals.

Compared with other methods, our identification process is time- and cost-saving, as the TPI-plex system is a single reaction multiplex system. Our research could contribute to the supplementation and correction of studbook records by identifying and verifying the pedigree relationships among captive individuals and could also play a positive role in promoting pedigree management and breeding control of tigers in captive institutions. It is of great significance to effectively avoid inbreeding in order to protect the genetic diversity of captive tigers.

## ACKNOWLEDGEMENTS

We thank Changsha Ecological Zoo and Sanzhen Tiger Park for providing the biological samples for the tigers. We thank Shenzhen Key Laboratory of Forensics. We thank Rihua Yang, Yixin Zhu, Songjiao Liu and Dami Jiang from the Key Laboratory of Forensics of BGI-Shenzhen for providing technical support and equipment debugging. We also thank Dong Li from Forensic Genomics International (FGI) of BGI-Shenzhen for providing us samples of human, sheep, chick, duck, dog, and rat.

### Funding

This work was supported by the Shenzhen Municipal Government of China (No. ZDSYS201507301424148). The funders had no role in study design, data collection and analysis, decision to publish, or preparation of the manuscript.

### Grant Disclosures

The following grant information was disclosed by the authors:
Shenzhen Municipal Government of China: ZDSYS201507301424148.

### Competing Interests

Xiao Zhao, Dongke Fu, Xuesong Hu and Shengjie Gao are employed by Forensics Genomics International (FGI), BGI-Shenzhen, Shenzhen, China. Xiao Zhao, Dongke Fu is employed by Shenzhen Key Laboratory of Forensics, BGI-Shenzhen, Shenzhen, China. Xiao Zhao, Chang Li, Dongke Fu, Xuesong Hu, Shengjie Gao, Huanming Yang and Bo Li are employed by BGI-Shenzhen, Shenzhen, China. Qiguan Qiu is employed by Changsha Ecological Zoo. Chang Li is employed by BGI-Qingdao, BGI-Shenzhen, Qingdao, China.

Changsha Sanzhen Tiger Park is a private zoo and Yugang Zhu is the owner of Changsha Sanzhen Tiger Park. Huanming Yang is employed by China National GeneBank, BGI-Shenzhen, Shenzhen, China. Mu Haofang is employed by Center of Forensic Sciences, BGI, Beijing, China. Runping Wang is employed by BGI Shaanxi Xixian new area Institute of Forensic Science, Xi'an, China.

## Author Contributions

- Xiao Zhao conceived and designed the experiments, performed the experiments, analyzed the data, prepared figures and/or tables, authored or reviewed drafts of the paper, and approved the final draft.
- Qiguan Qiu conceived and designed the experiments, authored or reviewed drafts of the paper, and approved the final draft.
- Chang Li conceived and designed the experiments, performed the experiments, analyzed the data, prepared figures and/or tables, authored or reviewed drafts of the paper, and approved the final draft.
- Dongke Fu conceived and designed the experiments, prepared figures and/or tables, and approved the final draft.
- Xuesong Hu analyzed the data, prepared figures and/or tables, and approved the final draft.
- Shengjie Gao analyzed the data, prepared figures and/or tables, and approved the final draft.
- Yugang Zhu conceived and designed the experiments, prepared figures and/or tables, and approved the final draft.
- Haofang Mu conceived and designed the experiments, authored or reviewed drafts of the paper, and approved the final draft.
- Runping Wang conceived and designed the experiments, authored or reviewed drafts of the paper, and approved the final draft.
- Huanming Yang conceived and designed the experiments, authored or reviewed drafts of the paper, and approved the final draft.
- Bo Li conceived and designed the experiments, authored or reviewed drafts of the paper, and approved the final draft.

## Animal Ethics

The following information was supplied relating to ethical approvals (i.e., approving body and any reference numbers):

The Institutional Review Board on Bioethics and Biosafety of BGI provided full approval for this research (FT 16084).

## Data Availability

The raw data of sequenced PCR amplification products from homozygotes are available in Table S4.

## Supplemental Information

Supplemental information for this article can be found online at http://dx.doi.org/10.7717/peerj.8939#supplemental-information.

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
