# Peer review of "Genome-based development of 15 microsatellite markers in fluorescent multiplexes for parentage testing in captive tigers"

_PeerJ, doi:10.7717/peerj.8939_

## Round 0.1 · original submission · Major Revisions

As commented by the Reviewers the ms suffers from serious flaws, including methodological. Please try to comply with the criticisms raised by the Reviewers

Reviewer 1 ·

Basic reporting

The article titled, "Genome-based development of 15 microsatellite markers in fluorescent multiplexes for parentage testing in tigers" describes new genetic markers mined by the authors, screened, validated, combined into a multiplex, and then used in actual parentage scenarios. The work done has been extensive and its impacts on the genetic diversity of the declining population of tigers broad.

The introduction starts with setting the historical landscape of why tigers have declined and how this has led to a loss in their genetic diversity through captive inbreeding. In the introduction it is evident that the written English language is not mastered by the authors. Sentence structure is of varied complexity and often lends to confusion of the purpose of the writing. For example, lines 59-61, 66-68. This confusion is increased in lines 58-89. I think the authors are trying to outline drawbacks of microsatellite detection methods, but this is not clear. It is recommended to separate information into smaller paragraphs that have a central focus and purpose then provide helpful transitions between paragraphs. Other lines that need clarification through proper use of English: 181-183, 202-203, 215-216, 250-251, 325-326.

Figure 1 helps visualize an overview of the methods and work that the author's completed, however, this manuscript is not a methods paper. Thus, it is in my opinion that dedicating a page and figure to the work flow is unnecessary as this information is gleaned from the methods in the paper.

Figure 2 is clear, and adds to the manuscript. The locations of the markers used in the multiplex are easy to visualize with this figure.

Figure 3 is informative, however, is better suited as supplemental information.
Figure 4 is unnecessary as this information can be gleaned from the electrophoretic data in Figure 7. In Figure 7, the loci are clearly labeled on the electropherograms and thus provides the information from Figure 4 and new information of Figure 7. Figure 4 should be eliminated as it is repetitive of information found in other figures.

The title and labeled for Figure 5 needs to be more descriptive.
Figure 8 is a creative way to show the concordance of sexing with this assay.

I think the information in Table 1 and 2 can be combined somehow to save space. They display similar information and with detailed table title and legend can be combined.

Information in Tables 5, 6, 7, 8, and 9 is extensive and makes it evident the extensive work done in this manuscript. However, much of the information should move to the Supplemental section. It is recommended that Tables 6, 8, and 9 as it shows data specific to this data set be moved to supplemental. The allele frequencies (Table 5) and population statistics (Table 7) are most needed as reference for a reader.

Experimental design

There has been extensive work done in preparation for this manuscript. It is not clear to the reader if other microsatellites have been identified and developed for tigers. The literature is not shared in this article. Upon a quick search, the following that have identified microsatellites and made multiplexes were found:
Williamson, J. E., Huebinger, R. M., Sommer, J. A., Louis Jr, E. E., & Barber, R. C. (2002). Development and cross‐species amplification of 18 microsatellite markers in the Sumatran tiger (Panthera tigris sumatrae). Molecular Ecology Notes, 2(2), 110-112.

ZHANG, Z. H., ZHANG, W. P., YUE, B. S., SHEN, F. J., Zhang, L., Hou, R., & ZHU, M. Y. (2006). Twelve polymorphic microsatellite loci for the South China tiger Panthera tigris amoyensis. Molecular Ecology Notes, 6(1), 24-26.

Zhang, Y., Li, D., Rao, L., Xiao, Q., & Liu, D. (2003). Identification of polymorphic microsatellite DNA loci and paternity testing of Amur tigers. Dong wu xue bao.[Acta zoologica Sinica], 49(1), 118-123.

Singh, A., Gaur, A., Shailaja, K., Bala, B. S., & Singh, L. (2004). A novel microsatellite (STR) marker for forensic identification of big cats in India. Forensic science international, 141(2-3), 143-147.

In fact, the citation by Xu, et al. (2005) is used in the manuscript to show that they measured the population is suffering from inbreeding (line 52-53), so why is this work that is under consideration needed? This begs to question, why re-mine the genome to find new microsatellites, why is the previous work not sufficient for determining paternity, how is this work novel? It appears to me that no research gap exists.

Methods sections 2.5, 2.9, and 2.10 are not sufficiently detailed to reproduce this work. For example, in 2.5, what kind of sequencing what performed? what is the depth of sequencing? In 2.9 and 2.10 the methods are not clear.

Validity of the findings

The conclusions are very short for such extensive work. The conclusions are concise and well stated, however, not linked back to the original research question or explain how it can address a need.

Additional comments

The English language and writing must be significantly improved throughout the manuscript. There are too many places to list lines. It is recommended that someone of native English speaking review and edit this manuscript.

Reviewer 2 ·

Basic reporting

English needs to be reviewed. Several relevant papers not included, background loses focus. See author comments.

Experimental design

For the most part the ED is good, although there are some confusing aspects with regards to true sample numbers tested, who is included in the database and why domestic cat is tested.

Validity of the findings

See my comments below.

Additional comments

The paper describes the development of 15 STR markers for tiger (P. tigris). This research is needed, but the paper needs major revision prior to publication. The English needs to be reviewed as there are some inconsistencies as well as spelling and grammatical errors. I have pointed out some of these errors in my comments below.

The structure of the paper is incorrect with the introduction consisting of methods and the main point of the development of test not being fully described or explored. The paper is far too long with large amounts of repetition that needs to be removed. The low number of individuals for validation (31), and the fact that several of them seem to be related, means that most of the frequency and statistical results are not reliable. This is also confusing, you state that you have 55 tigers (still too few for a database), and what is tested and what is included in the frequency tables is not clear. I don’t understand the point of including domestic cats, and so few if they are required. That being said, I think the work is important and by publishing the primer sequences other groups could expand the frequency tables and make the whole assay more reliable. I think that with significant revision the paper will be suitable for publication.

There are too many figures and tables.
Figure 1 can be removed. I don’t think this is necessary at all.
Figure 2 can be removed and put in the supplementary section if authors want.
Figure 5 can be removed.
Figure 6 can be removed as this is described in text. This data does not display well in this format.
Figure 7 – Your female tiger should be XX not YY, your male tiger is correct in that it has an XY. You need to switch your labels as they are incorrect. Based on these results there seem to be a lot of microvariants.
Figure 8 can be removed.
Table 2 I’m not clear on why domestic cats are included in this study.
Table 4 can be supplementary.
Table 5 the observed size can be removed from this table to condense.
Table 6 Supplementary.
Table 8 Supplementary.
Table 9 supplementary.

Line 19: suggest “captive breeding programs”.

Lines 21-28: This is a confusing section. I suggest something similar to:

Using this pipeline, we developed a new multiplex panel named TPI-plex (Tiger pedigree identification). By sequencing the entire tiger genome, 139,967 STR loci were identified, and 12.76% of these displayed three or more alleles among multiple individuals. A total of 214 candidate STRs were identified and screened with 31 unrelated individuals. Of these, 15 loci were chosen for inclusion in the multiplex. The mean allele number and mean expected heterozigosity (

·

Basic reporting

The authors have described the need for the study for ex-situ conservation of tigers and undertaken the all required analysis for validating their. (Details are in attached report).

Experimental design

Required experimental design and analysis are quite rigorous to validate the objective of the paper. Method are clearly described all the relevant information. (Details attached).

Validity of the findings

Proposed TPI plex would be useful in managing captive tiger populations more scientifcally.

Additional comments

Kindly see required specific changes needed in attached file.

---

## Round 0.2 · Major Revisions

I fear this new version is still not satisfactory . Please take into consideration the criticisms raised, with particular emphasis on the methodological questions, and controls.

Reviewer 1 ·

Basic reporting

No comment.

Experimental design

No comment.

Validity of the findings

No comment.

Reviewer 2 ·

Basic reporting

The English in this version is improved.

The references need to be checked to ensure the proper sentence structure is presented (capitalization especially).

Experimental design

I still have issues with the unrelated individuals - see below.

Validity of the findings

Again - reservations due to the relationship status of the tigers.

Additional comments

37-38 - IUCN stands for International Union for Conservation of Nature - please correct.

38-39 - please see Kitchener, Andrew C., et al. "A revised taxonomy of the Felidae: The final report of the Cat Classification Task Force of the IUCN Cat Specialist Group." Cat News (2017) for current internationally accepted classification for tiger subspecies. Note there are no long nine accepted subspecies, with balica and sumatrae included with sondaica. jacksoni is a nomen nudum and should not be used.

51-62 - Atkinson, Kirsty E., et al. "An assessment of the genetic diversity of the founders of the European captive population of Asian lion (Panthera leo leo), using microsatellite markers and studbook analysis." Mammalian Biology 88 (2018): 138-143 is a recent study demonstrating inbreeding effects in a related Panthera species.

81 - you developed the system, take out attempted.

93 - were sampled in this study

104 - Eppendorf.

130 - size ranges did not overlap

133-134 - even if you're not giving the final primer concentration for each primer, a range should be provided here.

139 - commas look strange.

150 - "The allele at each locus that was amplified from the homozygote was sent to Sangon Biotech.." - I don't understand this. Do you mean "A homozygote was selected at each locus and sent to..."

2.7 I'm still not clear on the unrelated individuals. Here you include 18 of the bengal tigers which you previously stated "96 - The 20 Bengal tigers from Changsha Ecological Zoo kept unclear relationships and 97 their partially known and probable genealogic information was provided by the zoo staff". If they are not clearly unrelated then they can't be included in these statistics.

242 - repeat numbers

260 - Take out the interestingly sentence - it is not interesting. You have 16 loci and 5-11 alleles, so SOME of your loci have to have the same number of alleles. It would be interesting if they all had different numbers of alleles.

306-307 - I have major issues here. You are using 31 tigers as your reference population that you claim are unrelated, however the numbers provided in this section as verified related, are also contained in the unrelated reference group. I think it would be fine if you removed T13 from your reference population - but this would require all frequencies and equations to be redone.

---

## Round 0.3 · Major Revisions

The revised manuscript still needs revision and further information on raw data is required. A table with absolute (observed, and desirably, expected) frequencies is mandatory in order to check some conclusions depending on the presence of recessive alleles and to explain the great number of loci not fitting HWE. I suggest modifying Table S6 accordingly.
It seems that no allelic ladder was used, which makes allele assignment more troublesome. If indeed it is the case, please refer explicitly and add a corresponding cautionary note.

Nomenclature is also to be checked and reviewed; I think your decision, when applied to complex motif loci is problematic to say the least and simply bp length would be more appropriate, less confusing and more fit to standardization.

Referencing needs extensive and careful revision.

NB: I add a commented version of the manuscript detailing my suggestions and required modifications.

---

## Round 0.4 · Minor Revisions

The ms still lacks some information essential to ensure the validity of conclusions and the reproducibility of results. My detailed comments and requests are in my answer to the rebuttal letter appended.

---

## Round 0.5 · accepted · Accept

Although I think nomenclature options taken are debatable, I think now readers can access data now provided and overcome the problem.